

# Disparities in COVID-19 incidence and fatality rates at high-altitude

Bilal Ahmed Abbasi[1], Neha Chanana[1], Tsering Palmo[1] and Qadar Pasha[1,2]

[1] CSIR-Institute of Genomics and Integrative Biology, Genomics and Molecular Medicine, Delhi, India
[2] Institute of Hypoxia Research, New Delhi, India

## ABSTRACT

**Background**. SARS-CoV-2 has affected every demography disproportionately, including even the native highland populations. Hypobaric-hypoxic settings at high-altitude (HA, >2,500 masl) present an extreme environment that impacts the survival of permanent residents, possibly including SARS-CoV-2. Conflicting hypotheses have been presented for COVID-19 incidence and fatality at HA.

**Objectives**. To evaluate protection or risk against COVID-19 incidence and fatality in humans under hypobaric-hypoxic environment of high-altitude (>2,501 masl).

**Methods**. Global COVID-19 data of March 2020-21, employed from official websites of the Indian Government, John Hopkins University, and Worldometer were clustered into 6 altitude categories. Clinical cofactors and comorbidities data were evaluated with COVID-19 incidence and fatality. Extensive comparisons and correlations using several statistical tools estimated the risk and protection.

**Results**. Of relevance, data analyses revealed four distinct responses, namely, partial risk, total risk, partial protection, and total protection from COVID-19 at high-altitude indicating a mixed baggage and complexity of the infection. Surprisingly, it included the countries within the same geographic region. Moreover, body mass index, hypertension, and diabetes correlated significantly with COVID-19 incidence and fatality rate ($P \leq 0.05$).

**Conclusions**. Varied patterns of protection and risk against COVID-19 incidence and fatality were observed among the high-altitude populations. It is though premature to generalize COVID-19 effects on any particular demography without further extensive studies.

## INTRODUCTION

COVID-19 (Coronavirus Disease 2019), a global pandemic caused by SARS-CoV-2, invariably targets respiratory and gastrointestinal tracts leading to loss of human life (*Wang et al., 2020*). SARS-CoV-2 evolves faster and escapes the host immune surveillance due to inadequate RNA polymerase proofreading capacity (*Ziebuhr, 2005*). SARS-CoV-2 employs angiotensin converting enzyme 2 (ACE2) to enter human epithelial cells. ACE2 is a vasorepressor and acts as an antagonist to its homologue ACE1 to maintain a dynamic equilibrium of the renin angiotensin aldosterone system (*Paul, Mehr & Kreutz, 2006*). Literature suggests that an individual's risk factors like age, gender, comorbidities, and

Corresponding author
Qadar Pasha,
qadarpasha@hotmail.com

various environmental factors are involved in the spread of the pandemic (*Sajadi et al., 2020*). Comorbid patients with hypertension, diabetes, chronic obstructive pulmonary disorder, and bronchitis remain highly vulnerable to the SARS-CoV-2 infection, resulting in more fatality (*Richardson et al., 2020*). Interestingly, altitude, one of the crucial environmental factors, has also gained a lot of attention.

High-altitude (> 2,501 metre-above-sea-level; masl) has always attracted humans; as a result, more than 140 million people have made it their permanent abode across several continents (*Penaloza & Arias-Stella, 2007*). High-altitude natives have acquired various advantageous physiological and morphological traits like high haemoglobin concentration, reduced arterial oxygen ($O_2$), increased resting ventilation, and minimally elevated pulmonary artery pressure to adapt to low $O_2$ environments (*West, 1991*; *Beall, 2007*). The prevailing high-altitude hypoxemic environment and COVID-19 infection could be challenging and further develop into dreadful complications due to $O_2$ severity, leading to respiratory failure and lung damage (*Gattinoni et al., 2020*). It, hence, generated lots of curiosity among the various high-altitude associated research groups in knowing whether altitude provides protection or contributes to risk in the high-altitude environment (Table S1). In the process, conflicting hypotheses have been reported during the past few months for the association of altitude with the COVID-19 fatality. A few groups have reported a higher risk of COVID-19 at high-altitude (*Luks & Swenson, 2020*; *Woolcott & Bergman, 2020*; *Millet et al., 2021*; *Cardenas, Valverde-Bruffau & Gonzales, 2021*; *Ortiz-Prado et al., 2021*), while others reported protection (*Segovia-Juarez, Castagnetto & Gonzales, 2020*; *Cano-Pérez et al., 2020*; *Stephens, Chernyavskiy & Bruns, 2021*; *Thomson et al., 2021*). Few of these reports also investigated the correlation of altitude with fatality, mortality, gender, age and diseases like diabetes and hypertension. Interestingly, mortality rates were reported higher in U.S. counties and Mexico municipalities located at ≥2,000 masl elevation *versus* those located <1,500 masl. The same team reported an association between altitude and COVID-19 mortality in men younger than 65 years (*Woolcott & Bergman, 2020*). Another study reported a lower incidence of COVID-19 infection at HA, but case fatality was not affected (*Segovia-Juarez, Castagnetto & Gonzales, 2020*). Comorbidities, especially diabetes, hypertension and other cardiopulmonary diseases, have been extensively investigated globally and associated with COVID-19 fatality and mortality; however, they have sparsely investigated for high-altitude populations (*Luks & Swenson, 2020*; *Leon-Abarca et al., 2021*). Importantly, each study was limited to a specific high-altitude country; none evaluated the COVID-19 effect involving several countries and continents to accurately picture the high-altitude regions (*Campos et al., 2021*; *Simbaña Rivera et al., 2022*). Because of this restriction, the sample sizes were also a limiting factor in these studies. We have been working in the high-altitude field for a long time; hence, this concept attracted us, too. However, instead of emphasizing a single country such as India or any other, we preferred to evaluate the available COVID-19 data of high-altitude populations from various geographies.

Here, for the first time, we compared the impact of COVID-19 in terms of incidence and fatality rates on major high-altitude regions, including the Himalayan ranges—India, Nepal, Bhutan, and China; the Atlas ranges—Morocco and African countries; the Southern

Rockies ranges—Mexico, Colorado; the Andes ranges—Colombia, Peru, Bolivia, Chile, and Argentina. Moreover, we emphasized the variable COVID-19 infection rate and severity in Indian states, majorly considering a global comparative perspective. Additionally, various clinical cofactors, including body mass index (BMI), blood pressure, haemoglobin, and major comorbidities such as hypertension and diabetes, were considered and correlated in an altitude-specific manner with respective incidence and fatality rates. Our extensive analyses relied on the openly accessible Indian and Global COVID-19 data.

## METHODS

### COVID-19 data collection

The records of COVID-19 confirmed cases and deaths were acquired from the Government of India website, (*Government of India, 2020*) and similar International COVID-19 data from the Johns Hopkins University (*Dong, Du & Gardner, 2020*) and the Worldometer website (*Worldometer, 2021*). Data of the Indian population residing in 36 states were obtained from the Census 2020 based on the 2011 report, while that of 185 international countries from (*City Polpulation, 2021*) and (*Statista, 2021*) (Table S2A).

### Altitude data collection

For the comparative analysis, the average altitude of each geography was acquired from the topographic website https://en-in.topographic-map.com/ (Table S2B).

### Stratification and comparisons of altitude-specific data

India ($n = 36$ states) and the world countries ($n = 185$) were stratified into two broader geographic categories: high-altitude and low-altitude, having an average height of $\geq 2,501$ masl and $<2,500$ masl, respectively. Moreover, the low-altitude region of each geography was further classified with an interval of 500 masl, starting from sea-level, 0–500 masl to intermediate heights of 501–1,000 masl, 1,001–1,500 masl, 1,501–2,000 masl, and 2,001–2,500 masl; data for these intermediate heights for all categories are available in Table S3.

### Inclusion and exclusion criteria

Only those countries whose data at cities/provinces/department-level were provided at the public domain were considered, and others were excluded. Missing data were deemed to be null.

### Evaluation of COVID-19 incidence, recovery and fatality rates

The incidence and fatality rates due to COVID-19 infection and recovery from the same were calculated using the following standard formulae, which helped homogenize the metrics.

$$\text{Incidence rate/IR (\%)} = \frac{\text{Total number of confirmed patients}}{\text{Total population}} \times 100$$

$$\text{Recovery rate/RR (\%)} = \frac{\text{Total number of cured/Discharged patients}}{\text{Total confirmed patients}} \times 100$$

$$\text{Fatality rate/FR (\%)} = \frac{\text{Total number of deceased patients}}{\text{Total confirmed patients}} \times 100$$

Sea-level, 0–500 masl was considered as a baseline reference for overall calculations and comparisons with various altitudes' groups such as 501–1,000 masl, 1,001–1,500 masl, 1,501–2,000 masl, 2,001–2,500 masl and ≥2,501 masl for each geography. Further, these analyses allowed us to compare the influence of altitude on the transmission and pathogenesis of COVID-19.

### Cofactors and comorbidities

Most pertinent cofactors and comorbidities were compared for India and the globe. Cofactors like BMI, blood pressure, and Hb concentration, while comorbidities like hypertension and diabetes were accessed from the official records and published datasets (Table S2C).

### Correlation analyses

The cofactors and comorbidities of India and the globe were correlated with incidence and fatality rates at the defined increasing altitudes, keeping 0–500 masl as a reference altitude. It determined the change in the association of parameters while moving from lowlands to highlands.

### Statistical analysis

For meaningful statistical inferences, IR and FR were calculated per 100 inhabitants (%). Data have been presented as mean and standard errors of the mean (SEMs). The differences in the mean of two groups were calculated using the unpaired two-tailed Student's $t$-test. The parameters, which were not in a normal distribution, were analysed by non-parametric Shapiro–Wilk's test to determine the suitability of parametric tests. All statistical tests were carried out using Sigma Plot version 12. Comparisons between the different altitudes were made against 0–500 masl. Statistical package for social sciences (SPSS 16.0) was employed to calculate the correlations. Pearson-partial correlation test calculated the correlation coefficient (R). Significance was maintained at $P \leq 0.05$.

## RESULTS

### Global COVID-19 status, incidence and fatality

SARS-CoV-2 infected 112.65 million individuals globally, caused approximately 2.49 million deaths, while 88.23 million patients recovered (Accessed: March 9-22, 2021). The United States, India, Brazil, Russia, and the United Kingdom had suffered the most with a total of 58.8 million infections. North America had the highest IR at 5.86%, with a relatively poor RR of 75.15%. South America exhibited the third-highest IR at 4.49%, with a reasonable RR of 89.65% (Table 1). Remarkably, Asia had the second-lowest IR at 0.56% and the highest RR of 93.40%. Country-wise, Andorra, Czechia, and the United States had the highest IR (Fig. S1A), Singapore and Bhutan had the highest RR, and Yemen had the highest FR (Figs. S1B–S1C). Of note, India accounted for 11.03 million infections in Asia, whereas 10.7 million patients recovered and 0.15 million succumbed to the virus. However,

**Table 1** Comparative analysis of continents for COVID-19 incidence, recovery, and fatality.

| Continent | Population in millions (Number of countries) | Incidence rate, % | Recovery rate, % | Fatality rate, % |
|---|---|---|---|---|
| Africa | 1363.04 (57) | 0.29 | 89.53 | 2.65 |
| Asia | 4633.37 (49) | 0.56 | 93.40 | 1.56 |
| Europe | 747.95 (48) | 4.88 | 70.80 | 2.35 |
| North America | 592.36 (39) | 5.86 | 75.15 | 2.28 |
| Oceania | 42.64 (12) | 0.12 | 64.96 | 2.08 |
| South America | 433.26 (14) | 4.49 | 89.65 | 2.58 |

**Notes.**

%, percentage; data retrieved from worldometer website https://www.worldometers.info/coronavirus/.

a controlled curve was observed from January 2021 to March 2021 for the United States, Brazil and India (Figs. S1D–S1E).

## COVID-19 incidence and fatality differentially associate with risk and protection for the global high-altitude population

Herein, high-altitude based exercises provided the most pertinent and exciting observations, differentiating the high-altitude regions into four distinct patterns, *i.e.,* partial risk, total risk, partial protection, and total protection. However, before presenting these patterns, India's COVID-19 results are presented.

### India

India, home to 1.35 billion people, representing a large chunk of the world population, had ∼11.03 million confirmed cases. Hence, studying the Indian COVID-19 pattern at the sea level or the highland is crucial. India's low-altitude population collectively showed an IR of 1.10 ±0.16%, which increased to 1.50 ±0.6% in the high-altitude population ($P > 0.05$). The incidence analyses with increasing altitude, starting from sea-level (0–501 masl), were in agreement with the average value (Fig. 1F; Table S3I). FR also depicted a similar trend, the low-altitude population collectively had an FR of 1.14 ± 0.11% that increased to 1.68 ± 0.18% at high-altitude ($P > 0.05$). Surprisingly, states at an altitude of 1,001–1,500 masl had an FR of 2.12 ± 0.51%. Overall, the patterns revealed that India's high-altitude population was at risk, both for COVID-19 incidence and fatality.

### The world trend
*Partial risk*

Partial risk is represented by a COVID-19 pattern of lower IR and higher FR with increasing altitude. African countries showed a lower IR of 0.32 ± 0.16% and higher FR of 4.68 ± 3.18% at 1,501–2,000 masl, whereas a higher IR of 0.79 ± 0.22% and lower FR of 2.84 ± 0.42% at 0-500masl (Fig. 1A). Argentina showed a lower IR of 3.20% and higher FR of 4.42% at ≥2,501 masl, whereas a higher IR of 7.33 ± 1.38% and lower FR of 1.68 ± 0.09% at 0–500 masl (Fig. 1B). Likewise, Chile showed a lower IR of 6.07% and higher FR of 2.01% at ≥2,501 masl, whereas a higher IR of 8.78 ± 1.93% and lower FR of 1.53 ± 0.39% at 0–500 masl (Fig. 1C). Lastly, Columbia also showed a lower IR of 2.43% and higher FR of 3.74% at ≥2,501 masl, whereas a higher IR of 3.04 ± 0.54% and lower FR of 2.78 ± 0.33% at

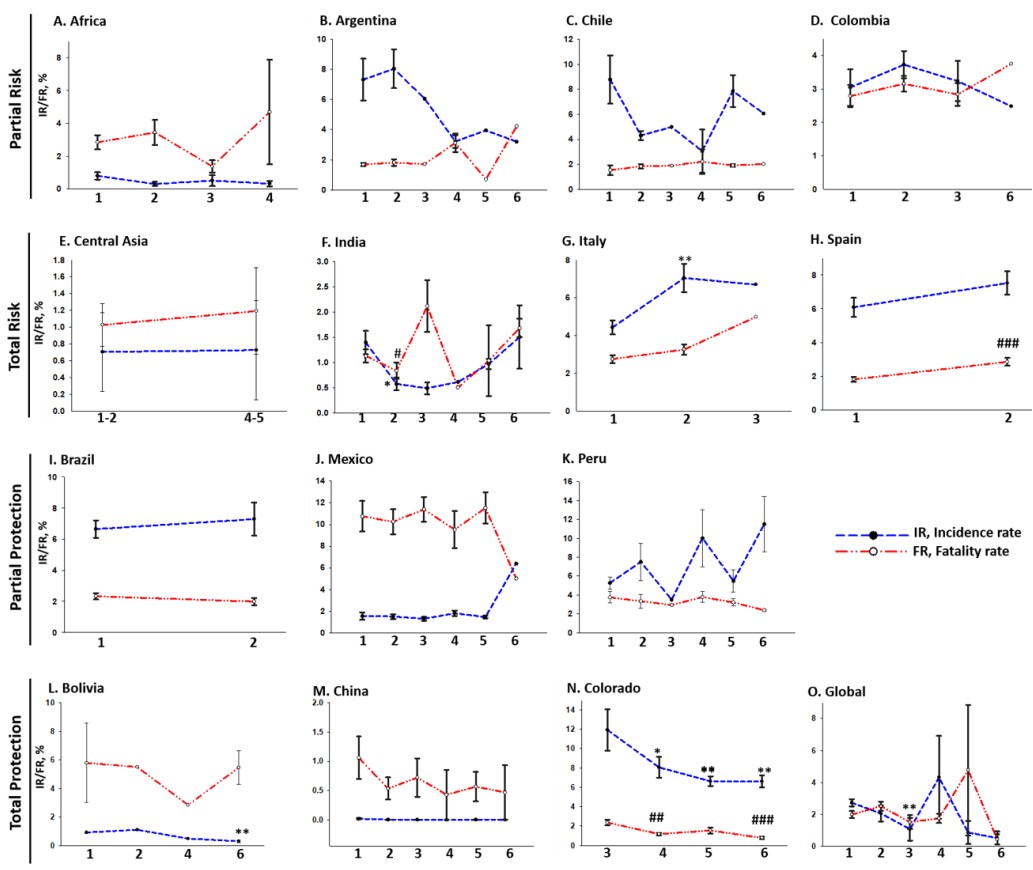

**Figure 1** **Varied pattern of COVID-19 IR and FR at different altitudes:** (A) Africa, (B) Argentina, (C) Chile, and (D) Columbia displayed partial risk to COVID-19 at high-altitude with a decreased incidence and increased fatality with an increase in altitude. On the other hand, (E) Central-Asia, (F) India, (G) Italy, and (H) Spain displayed a total risk of COVID-19 at high-altitude with an increased incidence and fatality with an increase in altitude. (I) Brazil, (J) Mexico, and (K) Peru, displayed partial protection against COVID-19 at high-altitude with an increased incidence and decreased fatality with an increase in altitude. On the other hand, (L) Bolivia, (M) China, (N) Colorado and (O) Global, displayed total protection against COVID-19 at high-altitude with a decreased incidence and decreased fatality with an increase in altitude. Statistical analysis was performed by two-tailed Student's t-test for the significance at the */#$P \leq 0.05$, **/##$P \leq 0.01$, and ***/###$P \leq 0.001$; ns-not significant, $P > 0.05$. * represents a comparison between Incidence rates, while # denotes Fatality rates. Altitude specificity is as follows, **1,** 0–500 masl; **2,** 501–1,000 masl; **3,** 1,001–1,500 masl; **4,** 1,501–2,000 masl; **5,** 2001–2,500 masl; **6,** $\geq 2,501$ masl.

0–500 masl (Fig. 1D). Results for the intermediate altitude categories are available in Table S3.

### Total risk

Both higher IR and FR represent total risk with increasing altitude. Central Asian countries showed an IR of $0.70 \pm 0.46\%$ and FR of $1.02 \pm 0.25\%$ at 501–1,000 masl, whereas the IR increased to 1.31% and FR to 1.70% at 1,501–2,000 masl (Fig. 1E). India showed the same pattern discussed in the previous section (Fig. 1F). Italy showed the highest IR of 6.70% and FR of 4.98% at 1,000–1,500 masl compared to the IR of $4.43 \pm 1.28\%$ and

FR of 2.75 ± 0.7% at 0–500 masl (Fig. 1G). Similarly, Spain also showed a higher IR of 7.53 ± 1.83% and FR of 2.85 ± 0.61% at 500–1,000 masl compared to IR of 6.09 ± 1.81% and FR of 1.81 ± 0.46% at 0–500 masl (Fig. 1H).

### Partial protection
Partial protection is identified by a pattern of higher IR and lower FR with increasing altitude. Brazil showed showed a higher IR of 7.28 ± 1.06% and a lower FR of 1.98 ± 0.23% at 501-1,000 masl compared to IR of 6.64 ± 0.56% and FR of 2.32 ± 0.20% at 0–500 masl (Fig. 1I). Similarly, Mexico showed a higher IR of 6.38% and a lower FR of 4.99% at ≥2,501 masl compared to IR of 1.53 ± 0.35% and FR of 10.75 ± 1.41% at 0–500 masl (Fig. 1J). Peru showed a higher IR of 11.50 ± 2.91% and lower FR of 2.39 ± 0.09% at ≥2,501 masl compared to IR of 5.27 ± 0.65% and FR of 3.74 ± 0.59% at 0–500 masl (Fig. 1K).

### Total protection
Total protection is identified by a pattern of lower IR and lower FR with increasing altitude. Bolivia showed an IR of 0.30 ± 0.08% and FR of 5.46 ± 0.08% at ≥2,501 masl compared to an IR of 0.88% and FR of 8.50% at 0–500 masl but with the exception of a few categories (Fig. 1L; Table S3). Data of China showed total protection at high-altitude, surprisingly with negligible values, as the IR of 3.41E-09 ± 0.001% and FR of 9.35E-07 ± 0.46% at ≥2,501 masl compared to an IR of 0.01 ± 0.009% and FR of 1.06 ± 0.36% at 0–500 masl (Fig. 1M). Colorado, an American county, was considered in this study due to its predominant high-altitude regions. It showed a lower IR of 6.59 ± 0.61% and FR of 0.77 ± 0.12% at ≥2,501 masl compared to an IR of 11.91 ± 2.12% and FR of 2.35 ± 0.27% at 1,001–1,500 masl (Fig. 1N). Interestingly, a similar pattern of lower IR and lower FR was revealed upon evaluating the global scenario (Fig. 1O).

## COVID-19 incidence and fatality rates correlated with cofactors and comorbidities
Here, various relevant clinical cofactors and comorbidities contributing to the virus infection and fatality were evaluated. The altitude-based evaluation provided interesting insights (Table S4). However, we opted for the standard and most crucial cofactors out of various cofactors for overall assessment.

### BMI
BMI at an average of ≥25.0 kg/m$^2$ decreased in Indians in the middle altitude categories, but it increased at high-altitude. It was 24.72 ± 2.10% at 0–500 masl and 26.95 ± 1.90% at ≥2,501 masl ($P < 0.05$, Fig. 2Ai). It correlated positively with COVID-19 IR in an altitude-specific manner ($P < 0.05$, Table S5A). BMI declined globally with increasing altitude; it ranged between 24.68 ± 0.30% at 0–500 masl and 20.60% at ≥2,501 masl (Fig. 2Bi) and correlated positively with COVID-19 IR ($P < 0.01$, Table S5B) and FR ($P > 0.05$) for all altitude categories.

### Blood pressure
India's elevated blood pressure of 180 mmHg/110 mmHg increased with increasing altitude. The per cent difference ranged between 0.81 ± 0.10% at 0–500 masl and 1.45 ± 0.45% at

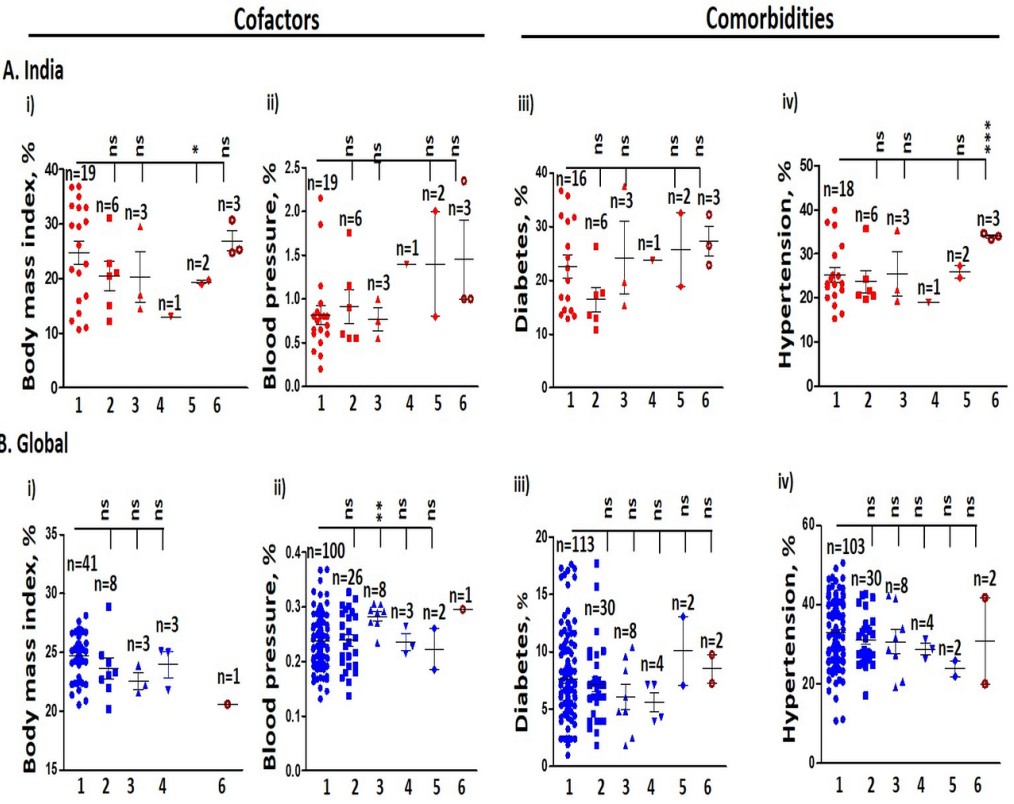

**Figure 2** **Cofactors and comorbidities in India and the globe.** (A) Indian population at increasing altitudes. (i) Body mass index. (ii) Blood pressure. (iii) Diabetes. (iv) Hypertension rates are higher at high-altitude with respect to altitude 0–501 masl. (B) Global population at increasing altitudes. (i) Body mass index is lower at high-altitude, while (ii) Blood pressure. (iii) Diabetes at high-altitude is higher compared to 0–500 masl. (iv) Prevalence of hypertension at high-altitude is lower compared to 0–501 masl. Statistical analysis was performed by two-tailed $t$-test for significance. *$P \leq 0.05$, **$P \leq 0.01$, and ***$P \leq 0.001$; ns-not significant, $P > 0.05$. 1. 0–500 masl; 2. 501–1,000 masl; 3. 1,001–1,500 masl; 4. 1,501–2,000 masl; 5. 2001–2,500 masl; 6. $\geq$ 2,501 masl. n represent the number of states and countries in India and globally, respectively.

$\geq$2,501 masl (Fig. 2Aii). Globally, blood pressure was 0.24% at 0–500 masl and 0.30% at $\geq$2,501 masl, which correlated positively with IR and FR ($P > 0.05$, Fig. 2Bii).

## Diabetes

The average per cent prevalence of diabetes in India elevated with increasing altitudes. It ranged between 22.67 $\pm$ 2.14% at 0–500 masl and 27.23 $\pm$ 2.73% at $\geq$2,501 masl (Fig. 2Aiii). It correlated positively with COVID-19 IR for all the altitude categories ($P < 0.05$, Table S5A) and also with FR. Similarly, the prevalence of diabetes around the globe followed the Indian trend; the average value of diabetes prevalence ranged between 7.66 $\pm$ 0.36% at 0–500 masl and 8.50 $\pm$ 1.24% at $\geq$2,501 masl (Fig. 2Biii). Further, it correlated positively with IR but inadequately with FR ($P > 0.05$).

## Hypertension

Hypertension status for India and the globe are clustered in Fig. 2A. Hypertension prevalence in India was 25.22 ± 1.66% at 0–500 masl and 33.98 ± 0.38% at ≥2,501 masl ($P < 0.05$). It correlated positively with IR for all the altitude categories (Table S5A), but did not correlate significantly with FR when the altitude was treated as a control variable. With increasing altitude, the global prevalence of hypertension was 32.87 ± 0.84% at 0–500 masl and 30.95 ± 10.90% for ≥2,501 masl (Fig. 2Biv). It correlated positively with all altitude categories ($P < 0.001$, Table S5B), however the correlation did not reach significance with FR ($P > 0.05$).

## DISCUSSION

Our epidemiological analyses suggest that any direct association of COVID-19 with altitude could oversimplify a complex relationship. At the onset, it seems the virus is equally effective both in normoxic and hypoxic environments. Several nations at sea-level or normoxic environments suffered significant fatality due to COVID-19; hence, it has been conceptualized by various investigators that the hypobaric-hypoxia environment may protect from fatality if not from the incidence rate. The most striking findings of our evaluation are four distinct high-altitude associated patterns: partial risk, total risk, partial protection, and total protection against COVID-19. These findings of variable SARS-CoV-2 infection contest the reports favouring protection for high-altitude population (*Segovia-Juarez, Castagnetto & Gonzales, 2020*; *Cano-Pérez et al., 2020*).

Our findings are exciting and perplexing, mainly because countries within the same geographic region show different COVID-19 incidence and fatality patterns. The environment is undoubtedly similar, but the COVID-19 effect is not, perhaps, highlighting the complexity of infection. Among the South American countries that have a hypoxic environment, Argentina, Chile, and Bolivia fell under partial risk, but none of the countries fell into the total risk category. Brazil, Mexico and Peru fell under partial protection, while Bolivia was under total protection. These findings also pointed to the effect of lockdown stringency, affecting COVID-19 IR and FR (*Thomson et al., 2021*). Moreover, these findings were surprising and contrasting for the IR and FR of several European countries and the USA, which suffered greatly. It vaguely pointed to the contributions, in addition to the environment, of other factors, including the travel pattern and socio-economic status, but it was beyond our control to analyze these two. Nonetheless, we could access and analyze the other equally relevant data. Broadly, these findings pointed to a modest COVID-19 effect in this geographic region.

In our evaluations, cofactors such as BMI and blood pressure and comorbidities such as diabetes and hypertension increased in an altitude-specific manner for India, similar to the COVID-19 IR and FR patterns. BMI and hypertension showed a significant positive correlation with IR, but a similar trend could not be observed with FR; the reason could be the lesser number of states and also the significantly decreased population size. Diabetes, IR and FR were proportionally correlated to each other. Globally, BMI and blood pressure showed a significant positive correlation with COVID-19 IR and FR. An increased BMI is

an impending risk of developing obesity, and COVID-19 patients with obesity saw greater hospitalisation (*Zhou et al., 2020*). Obesity, in turn, accounts for higher risk of developing diabetes (*Ko et al., 2021*). Diabetes associated positively with global IR but not significantly with FR. Similarly, hypertension also correlated positively with COVID-19 IR for India and the globe. A meta-analysis study showed an almost 2.5-fold increased risk of COVID-19 in the presence of hypertension, while the mortality risk depicted the same trend with an odds ratio of 2.42 (*Lippi, Wong & Henry, 2020*). Thus, our observation indicated that BMI, blood pressure, diabetes and hypertension contributed significantly to the risk of COVID-19. Our observations align well with studies performed directly with patients hospitalized with COVID-19 who had clinical conditions such as obesity, high blood pressure, and diabetes (*Algoblan, Alalfi & Khan, 2014*). Moreover, the increased expression of ACE2 and ACE1 in type1 and type2 diabetes, high blood pressure, and hypertension is often treated with ACE inhibitors and angiotensin II type-I receptor blockers (*Tseng, Yang & Lu, 2020*; *Guan et al., 2020*; *Zhang et al., 2020*). In case of COVID-19 patients, however, a cautious approach is needed. Comorbidities, in the presence of COVID-19 infection, exaggerate the vascular system, causing more vasoconstriction that has severe consequences leading to multi-organ diseases (*Zhang et al., 2020*), thereby multiplying the severity in the case of COVID-19 that attacks the respiratory system. Hence, people suffering at sea level (normoxia) or highland (hypoxia) probably will have similar clinical conditions and severity.

We desire to highlight a few additional cofactors that are also pertinent to COVID-19 incidence and fatality. First and foremost is Hb, whose concentration in the high-altitude environment (>2,501 masl) has been extensively studied. Because this molecule directly correlates with oxygen saturation. The high-altitude dwellers are known to have normal to significantly increased concentrations of Hb. Based on the meta-analysis of Hb data, altitude-specific Hb levels increased the most in regions falling under the total protection category, followed by the partial protection/risk category, and the least Hb increase in regions falling under the total risk category. This aligns with the increased COVID-19 severity associated with decreased Hb levels (*Anai et al., 2021*). The other risk factor is old age.

The geriatric population with hypertension, diabetes, and other comorbidities suffered maximum fatality, which substantiated that old age, altitude, comorbidities and COVID-19 proved fatal to human life (*Dhama et al., 2020*; *Richardson et al., 2020*; *Singh et al., 2020*). Likewise, it was difficult to overlook the genetic distribution of the high-altitude population and the COVID-19 incidence. Understandably, the genetic distribution among the various high-altitude ethnicities differed significantly, which might also contribute to varied COVID-19 incidence rates. Amongst the most studied genes are *EPAS1*, *EGLN1*, *ACE*, *NOS3*, *APLN*, *ET1*, and *ARNT2* (*Beall, 2000*; *Simonson et al., 2012*; *Mishra et al., 2015*).

During the COVID-19 pandemic, treatment has been another major issue. Notable drugs like arbidol, camostat mesylate, repurposed drugs such as remdesivir, lopinavir/ritonavir, hydroxychloroquine, chloroquine, and the vaccines were prescribed but not without conflicts (*Chanana et al., 2020*; *Forni & Mantovani, 2021*; *Wouters et al., 2021*; *Abbasi et al., 2022*). Due to the similarity of hypoxia-induced symptoms in COVID-19 and high-altitude disorders such as acute mountain sickness and high-altitude pulmonary edema, some

initial studies had also suggested using drugs like nifedipine, phosphodiesterase inhibitors, erythropoietin, and dexamethasone (*Soliz et al., 2020*; *Solaimanzadeh, 2019*). However, experts rejected the claim and debunked the analogy of high-altitude illnesses with the COVID-19 (*Luks et al., 2020*; *Strapazzon et al., 2020*; *Archer, Sharp & Weir, 2020*; *Berger, Hackett & Bärtsch, 2020*). Thus the best option remained the vaccines that eventually overtook the market, and a good number of the world population, including the high-altitude, have been vaccinated.

## CONCLUSION

To conclude, our study highlighted four altitude-specific patterns for the first time *i.e.*, partial risk, total risk, partial protection and total protection, concerning COVID-19 incidence and fatality and their association with pre-existing cofactors and comorbidities at the global level. These findings are vital to the healthcare of the permanent residents of high-altitude and could even be significant to the overall global population for the effective management of COVID-19.

## PERSPECTIVE AND LIMITATIONS

Since SARS-CoV-2 had turned into a pandemic, scientists and policymakers raced to control COVID-19. Individuals with clinical symptoms were considered confirmed patients; the actual number will never be known. Importantly, under-reporting and disparity in information at high-altitude *versus* low-altitude were among major issues; in fact, it applied to the globe. Due to the unavailability of individual-level data and a common repository, we could not determine the correlation between one or the other cofactors or confoundings and the COVID-19 IR/FR weightage. To tackle it, data sharing needs to be as open as possible and should be analyzed carefully while drafting and implementing strategies for any particular geography. We are aware that mechanisms exterior to the scope of this study may contribute to the uneven rate at high-altitude. However, a perspective has been presented. Moreover, our observation of different patterns of risk and protection at high-altitude revealed that it might be a bit premature to make a sweeping statement about a recently discovered pandemic. However, further experiments are necessary to corroborate the data and findings to avoid the risks to the world population.

## ACKNOWLEDGEMENTS

We acknowledge open-source websites like Johns Hopkins University, Worldometer, Ourworldindata, WHO, and official Indian government websites for accessing the data.

### Funding

We received the support of the Cardiovascular Medical Research and Education, Philadelphia, USA (IGIB grant code CLP0032; BAA, NC); CSIR-UGC, New Delhi (ref

(21/06/2015(i)EU-V; T.P.); and Indian Council of Medical Research, New Delhi, India (ICMR No. 74/6/2015- Pers. EMS; QP). No additional external funding was received for this study. The funders had no role in study design, data collection and analysis, decision to publish, or preparation of the manuscript.

### Grant Disclosures

The following grant information was disclosed by the authors:
Cardiovascular Medical Research and Education, Philadelphia, USA (IGIB grant code CLP0032; BAA, NC).
CSIR-UGC, New Delhi (ref (21/06/2015(i)EU-V; T.P.).
Indian Council of Medical Research, New Delhi, India (ICMR No. 74/6/2015- Pers. EMS; QP).

### Competing Interests

The authors declare there are no competing interests.

### Author Contributions

- Bilal Ahmed Abbasi conceived and designed the experiments, performed the experiments, analyzed the data, prepared figures and/or tables, authored or reviewed drafts of the article, and approved the final draft.
- Neha Chanana performed the experiments, analyzed the data, prepared figures and/or tables, authored or reviewed drafts of the article, and approved the final draft.
- Tsering Palmo performed the experiments, analyzed the data, prepared figures and/or tables, authored or reviewed drafts of the article, and approved the final draft.
- Qadar Pasha conceived and designed the experiments, performed the experiments, analyzed the data, authored or reviewed drafts of the article, and approved the final draft.

### Data Availability

  The raw measurements are available in the Supplemental Files.

### Supplemental Information

Supplemental information for this article can be found online at http://dx.doi.org/10.7717/peerj.14473#supplemental-information.

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
