# Peer review of "Disparities in COVID-19 incidence and fatality rates at high-altitude"

_PeerJ, doi:10.7717/peerj.14473_

## Round 0.1 · original submission · Major Revisions

Thank you for submitting the manuscript. Please address all the comments from reviewers.

Reviewer 1 ·

Basic reporting

-previous studies have discussed altitude residence and covid. Most recent articles include but not limited too:
Simbaña-Rivera K, Jaramillo PRM, Silva JVV, Gómez-Barreno L, Campoverde ABV, Novillo Cevallos JF, Guanoquiza WEA, Guevara SLC, Castro LGI, Puerta NAM, Guayta Valladares AW, Lister A, Ortiz-Prado E. High-altitude is associated with better short-term survival in critically ill COVID-19 patients admitted to the ICU. PLoS One. 2022 Mar 31;17(3):e0262423. doi: 10.1371/journal.pone.0262423. PMID: 35358185; PMCID: PMC8970356.

Cardenas L, Valverde-Bruffau V, Gonzales GF. Altitude does not protect against SARS-CoV-2 infections and mortality due to COVID-19. Physiol Rep. 2021 Jun;9(11):e14922. doi: 10.14814/phy2.14922. PMID: 34110706; PMCID: PMC8191172.

Campos, A., Scheveck, B., Parikh, J. et al. Effect of altitude on COVID-19 mortality in Ecuador: an ecological study. BMC Public Health 21, 2079 (2021). https://doi.org/10.1186/s12889-021-12162-0

Millet GP, Debevec T, Brocherie F, Burtscher M, Burtscher J. Altitude and COVID-19: Friend or foe? A narrative review. Physiol Rep. 2021 Jan;8(24):e14615. doi: 10.14814/phy2.14615. PMID: 33340275; PMCID: PMC7749581.

it is suggested to provide a summary of previous literatures' findings in introduction to provide a big picture of current research progress

-it is suggested to address the correlation between previously reported data on altitude/hypoxia/altitude disease and COVID cases

Experimental design

-How are cofactors/comorbidities including previous history of respiratory systems, age, which potentially contribute to fatality assessed?

Validity of the findings

clear layout of data and statistically sound

Additional comments

-some grammatical errors including run-on sentences. Please revise English writing of the manuscript

Reviewer 2 ·

Basic reporting

The authors presented analysis of SARS-CoV-2 and how it affected demography disproportionately and also described the impact in hypobaric-hypoxic settings at high altitude. They grouped altitude into six different categories and performed correlation analysis for cofactors and comorbidities (hypertension and diabetes). Different groups were established based on Incident rate and Recovery rate factors. Comparisons of differences across groups were carried out using unpaired two-tailed student t-test.


1. The methods section (line 117) it has been mentioned that ".. were obtained from the Census 2020 based on the 2011 report". Did the authors use the 2011 report for this analysis? Can you please help explain.


2. The English language should be improved to ensure that an international audience can clearly understand your text. Some examples where the language could be improved include lines

line 84 - We "being working" in the high-altitude field for a long time, this concept attracted us, too.

Please consider changing "being" working to "been" working


line 187

SARS-CoV-2 had infected 112.65 million individuals globally, caused approximately 2.49 million deaths, while 88.23 million patients recovered (Accessed: March 9-22, 2021).

please change causing to caused.


line 207
India’s low-altitude population collectively showed an IR of 1.10±0.16% that increased to 1.50±0.6% in the high-altitude population

Please consider changing "that increased" to "which increased"



line 381

These findings are vital to healthcare of permanent residence of high altitude

Please consider changing residence to resident.

Experimental design

The research question is well defined, relevant and meaningful. Evaluation of COVID-19 incidence rates, recovery and fatality rates . Statistical analysis was done using unpaired two-tailed student's t-test. Non normal parameters were evaluated using non parametric sharpie wills test. The analysis was sound.

Validity of the findings

I commend authors for their time to collect and aggregate data from various sources. Underlying data used for this analysis was not provided. Conclusions were well stated. However, we could observe that there is a lack of samples / uneven rate in high altitude. The box plot in figure 2 shows very low samples for clusters 5 &6 across India and Global populations for both cofactors and comorbidities. Hence strong statements cannot be made about the validity of the analysis. If additional data can be gathered for these sub groups, that may help bolster this paper significantly.

Reviewer 3 ·

Basic reporting

No comment

Experimental design

The authors separately assessed the impact of altitudes and the impact of confounding factors (cofactors & combabilities) on the pattern of Covid-19 IR and FR. It appears that the impact of altitudes may partially attribute to the confounding, as the authors indicated in discussion. Methods estimating IR/FR while accounting for confounding may help better assess the relationship of altitude and COVID, accounting for the confounding factors such as BMI. For example, regression-based methods or CMH estimates maybe used. The authors can also consider weighted estimates of COVID IR/FR by BMI percentiles.

Validity of the findings

If methods accounting for confounding are not feasible, please add discussions to that and include it as one of the limitations.

Additional comments

Minor comment on Figure 1: both "#" and "*" were used, but the figure description only included "*". Please make update accordingly.

---

## Round 0.2 · accepted · Accept

Thank you so much for addressing all reviewers' comments.

Reviewer 1 ·

Basic reporting

The authors addressed all my concerns. The manuscript is satisfactory and can be accepted

Experimental design

Not applicable

Validity of the findings

Not applicable

Reviewer 2 ·

Basic reporting

no comment

Experimental design

no comment

Validity of the findings

no comment

Reviewer 3 ·

Basic reporting

No comment

Experimental design

No comment

Validity of the findings

No comment

Additional comments

The authors appropriately addressed my comments. I don’t have any additional ones. Thanks!